# Predicting the Early Response to Neoadjuvant Therapy with Breast MR Morphological, Functional and Relaxometry Features—A Pilot Study

**DOI:** 10.3390/cancers14235866

**Published:** 2022-11-28

**Authors:** Roxana Pintican, Radu Fechete, Bianca Boca, Madalina Cambrea, Tiberiu Leonte, Oana Camuescu, Diana Gherman, Ioana Bene, Larisa Dorina Ciule, Cristiana Augusta Ciortea, Sorin Marian Dudea, Anca Ileana Ciurea

**Affiliations:** 1Department of Radiology, “Iuliu Hatieganu” University of Medicine and Pharmacy, 400012 Cluj-Napoca, Romania; 2Department of Radiology, Emergency County Hospital, 400347 Cluj-Napoca, Romania; 3Department of MRI Phisics, Technical Univeristy, 400114 Cluj-Napoca, Romania; 4Department of Oncology, Emergency County Hospital, 400347 Cluj-Napoca, Romania

**Keywords:** NAT, neoadjuvant, predicting, MRI, relaxometry

## Abstract

**Simple Summary:**

Patients with non-metastatic breast cancer may achieve partial or complete response after neoadjuvant treatment. Early response to treatment after two cycles of chemotherapy has been associated with complete pathological response, longer disease-free interval and better overall survival. We aimed to investigate the role of morphologic, functional and relaxometry MR characteristics in predicting early response to neoadjuvant treatment. Tumor size, estrogen and progesterone receptors, HER2 status, Ki67%, tumor margins, enhancement type or ADC values were not associated with response. The T2 min relaxometry parameter and nodal stage were associated with early response, achieving an AUC of 0.826 [95% CI: 0.66–0.90, *p*-value < 0.001].

**Abstract:**

**Aim**: To evaluate the role of MR relaxometry and derived proton density analysis in the prediction of early treatment response after two cycles of neoadjuvant therapy (NAT), in patients with breast cancer. **Methods**: This was a prospective study that included 59 patients with breast cancer, who underwent breast MRI prior (MRI1) and after two cycles of NAT (MRI2). The MRI1 included a sequential acquisition with five different TE’s (50, 100, 150, 200 and 250 ms) and a TR of 5000 ms. Post-processing was used to obtain the T2 relaxometry map from the MR acquisition. The tumor was delineated and seven relaxometry and proton density parameters were extracted. Additional histopathology data, T2 features and ADC were included. The response to NAT was reported based on the MRI2 as responders: partial response (>30% decreased size) and complete response (no visible tumor stable disease (SD); and non-responders: stable disease or progression (>20% increased size). Statistics was done using Medcalc software. **Results**: There were 50 (79.3%) patients with response and 13 (20.7%) non-responders to NAT. Age, histologic type, “in situ” component, tumor grade, estrogen and progesterone receptors, ki67% proliferation index and HER2 status were not associated with NAT response (all *p* > 0.05). The nodal status (N) 0 was associated with early response, while N2 was associated with non-response (*p* = 0.005). The tumor (T) and metastatic (M) stage were not statistically significant associated with response (*p* > 0.05). The margins, size and ADC values were not associated with NAT response (*p*-value > 0.05). The T2 min relaxometry value was associated with response (*p* = 0.017); a cut-off value of 53.58 obtained 86% sensitivity (95% CI 73.3–94.2), 69.23 specificity (95% CI 38.6–90.9), with an AUC = 0.715 (*p* = 0.038). The combined model (T2 min and N stage) achieved an AUC of 0.826 [95% CI: 0.66–0.90, *p*-value < 0.001]. **Conclusions**: MR relaxometry may be a useful tool in predicting early treatment response to NAT in breast cancer patients.

## 1. Introduction

The current available treatment options for breast cancer patients include loco-re-gional (surgery or radiation) and systemic therapy (chemotherapy or endocrine therapy), depending on the tumor size, immunocytochemistry characteristics and cancer stage. Systemic therapy increases the overall survival and disease-free survival of patients, regardless if chemo- or endocrine therapy is given prior to surgery (neoadjuvant) or after the surgery (adjuvant) [1].

Neoadjuvant therapy is currently the standard of care for selected cases of early breast cancer patients axilla (eligible for breast conservation) and all locally advanced cancers (involving breast and axilla) and inflammatory breast cancers, as it has a number of potential benefits: may shrink the tumor and allow breast-conserving therapy, down-stage axillary lymph node disease and avoid axillary dissection and permits an “in vivo” evaluation of the tumor response to the treatment with prognostic information [2,3].

Neoadjuvant therapy is able to provide achievement of pathological complete re-sponse (pCR) associated with improved disease-free and overall survival of patients [4]. However, pCR is obtained in only 30% of the patients after completing the therapy, making it desirable to identify and predict as early as possible patients who will not benefit from neoadjuvant treatments.

MRI is the most accurate method for assessing tumor response and residual disease, when compared to clinical breast examination or other imaging techniques such as mammography or ultrasound [5,6]. Dynamic contrast-enhanced (DCE) MRI, or diffusion weighted imaging (DWI) with ADC values, were reported as having the potential for improving the prediction of the treatment response [7,8,9,10,11,12].

New approaches, such as machine learning with radiomic data based on MRI or circulating tumor DNA, have also been investigated to predict treatment response [13,14,15]. However, these techniques may lack reproducibility and can be time-consuming.

Breast MR relaxometry is a quantitative MR technique which has recently been re-ported as a reliable and reproductible method, allowing repeatable measurements across scan intervals and scanners, with possible clinical implications [16]. Furthermore, tissue-specific relaxometry data may differentiate between benign and malignant breast masses [17].

Our study aims to investigate the role of breast relaxometry as a possible imaging-marker in predicting the early response to therapy, after two cycles of neoadjuvant treatment, relying of the relaxometry characteristics of the tumors. MR relaxometry’s independent role and together with other features, such as size, MR morphologic features, MR ADC values and histology characteristics, will be included in the analysis.

## 2. Methods

### 2.1. Patients

This IRB-approved prospective study included patients with breast cancer who were scheduled for neoadjuvant therapy (NAT) and underwent breast MRI prior to and after two cycles of NAT. All patients presented to our institution between March 2021 to June 2022, and agreed to sign a written consent.

The exclusion criteria were: patients with per-primam surgery or patients who re-fused the neoadjuvant therapy, patients with incomplete MRI examination or histology reports.

Patient data including age, TNM stage and complete histology reports (tumor grade, estrogen/progesterone receptor status, ki-67% proliferation index and HER2 status) were also analyzed within the study.

The neoadjuvant therapy (NAT) consisted of chemotherapy (NACT) or endocrine therapy (NAHT) and is further mentioned simply as NAT. Complete therapeutic schemes are available in the.

### 2.2. MRI Relaxometry Sequences and Processing

For each patient, the 1.5T MRI examination included 5 different T2WI relaxometry in-house designed sequences with TE (TE1 = 50 ms, TE2 = 100 ms, TE3 = 150 ms, TE 4 = 200 ms and TE5 = 250 ms) and TR = 2000 ms; additionally, DWI apparent diffusion coefficient (ADC) with 3 b-values (50, 500, 800) and DCE (dynamic contrast-enhancement, 3D VIBRANT) sequences were acquired before NAT (MRI1) and after 2 cycles of NAT (MRI2).

The relaxometry images were exported and processed using a previously validated software to obtain the T2 relaxometry map. All breast tumors were delineated on the T2 relaxometry map by a breast radiologist with 4 years of experience, and 200 relaxometry data were automatically extracted for each tumor (Figure 1).

In addition, the T2 relaxometry processing derived the hydrogen proton density map (1H map), corresponding to each analyzed breast mass (Figure 2).

For patients having multifocal/multicentric disease with similar histology, the largest tumor was selected for the study. In cases with bilateral breast carcinoma, one tumor per breast was included (1 patient = 2 tumors).

MRI features, such as location (upper and lower, outer and inner quadrants), size, margins (circumscribed or non-circumscribed) and ADC values (with a standardized ROI of 0.2 mm^2^ obtained from the darkest area corresponding to the tumor), were included.

The early response assessed by MRI2 was divided into two groups: responders and non-responders. The responders group consisted of patients with complete response (CR = no disease is seen as mass enhancement/restricted diffusion) and partial response (PR = reduction in size ≥ 30%). The non-responder group included patients with progressive disease (PD = increase in size ≥ 20% or new lesion) and stable disease (SD = reduction is size < 30% or increase in size < 20%).

### 2.3. Statistics

Statistical analyses were performed using MedCalc software (version 19.2.6, Ostend, Belgium). For analyses associations between clinicopathological data and MRI features, the Chi-square or Fisher’s exact test was used.

The differences in age, ADC and relaxometry values between the two groups (responders and non-responders) were evaluated using independent Student’s *t*-test or nonparametric Mann–Whitney U test, in case of non-normally distributed data.

Receiver operating characteristic (ROC) curve analysis was conducted and area un-der the curve (AUC), sensitivity (Se), specificity (Sp), positive predictive value (PPV) and negative predictive value (NPV) were calculated to evaluate the diagnostic performance of the individual ADC first-order features for the prediction of metachronous metastases. The optimal cut-off value was chosen according to the Youden index. Using binary logistic regression (enter method), the authors created a combined model, which was also evaluated using ROC curve analysis. ROC curves were compared using the method developed by DeLong et al. A *p*-value of <0.05 was considered statistically significant.

## 3. Results

A total number of 63 breast tumors from 59 patients were included in the study, with a median age of 58 years (between 31–74 years), who underwent NAT as follows: 38 (64.4%) NACT and 21 (35.6%) NAHT (Figure 3).

Histology revealed invasive ductal carcinoma of no special type (IDC-NST) in 44 (88%) of the responders and 11 (84%) of the non-responders (*p*-value = 0.52). There was no association between the other special histology (such as lobular carcinoma, mucinous, medullary or papillary carcinoma) or an “in situ” component between responders and non-responders (all *p*-values > 0.05).

There were three (5%) responder patients with bilateral breast cancer and one (7%) non-responder patient, with no association between the multifocal/multicentric carcinoma between the two groups (all *p*-values > 0.05).

Size, histologic tumor grade, immunocytochemistry characteristics (ER, PR and HER2 status) and the ki67% proliferation index were not associated with NAT response (all *p*-values > 0.05).

The majority of the breast tumors were tumor (T) stage 2 (between 2–5 cm), with no statistically significant association between the two groups (*p*-value = 0.96).

The lymph node (N) 1 stage was associated with responders, while N 2 was associ-ated with non-responders (*p*-value = 0.005) (Table 1).

In the MRI, 100% of the breast tumors were depicted as masses on DCE sequences, with 13 (26%%) responders and 2 (15%) non-responders accompanied by non-mass enhancement on MRI (*p*-value = 0.641).

ADC values were not statistically significantly associated with responders or non-responders (*p*-value = 0.872).

MR relaxometry values obtained after post-processing for Pixel NR, T2max, T2av and derived proton density values of 1Hmax, 1Hmin and 1H av were not associated with responders or not responders (all *p*-values > 0.05) (Table 2 and Figure 4).

The T2min relaxometry values were associated with responders (*p*-value = 0.017). For the cut-off value of 53.58, the T2min parameter obtained 86% sensitivity (95% CI: 73.3–94.2), 69.23 specificity (95% CI: 38.6–90.9) and AUC of 0.715 (95% CI: 0.588–0.822) in pre-dicting early response to NAT (*p* = 0.038) (Figure 5).

Using logistic regression, we incorporated T2min values and N stage into a combined model, which achieved a higher AUC of 0.826 [95% CI: 0.66–0.90, *p*-value < 0.001] with a sensitivity of 88% and a specificity of 69.2% (Figure 6). However, this was not significantly different from the AUC of T2min (*p*-value = 0.34) (Figure 7).

## 4. Discussion

In the current study, we evaluated the role of breast relaxometry as a possible imag-ing-marker in predicting early response to treatment, after two cycles of NAT. We found that the T2min relaxometry parameter and a combined model (T2min and N stage) can predict early response after NAT.

In early breast cancer, complete or partial clinical response can be achieved with NAT, which may increase the breast-conserving surgery rate and might be more likely to eradicate micrometastatic disease [18]. Patients who achieved pathological complete response (pCR) after NAT had significantly better disease-free survival (DFS) and overall survival (OS) than those without pCR [19,20]. However, there are several NAT disadvantages, such as (1) it alters staging, (2) treatment delay of non-responder cases, (3) residual intra-ductal component may be left behind after breast conserving surgery, and (4) there are some cases of over-treatment [21].

Therefore, there is an increased interest in predicting early clinical response after two cycles of NAT, which further was reported to be associated with the pCR rate [22]. Early identification of clinical response in NAT therapy may help us to select the responders and to provide an opportunity to obtain alternative therapy for non-responders.

Ultrasound (US), MRI and F-FDG PET/CT have been evaluated for early response in breast cancer NAT. Diagnostic accuracy of US showed moderate results with different sensitivity and specificity depending on the BC molecular subtype [22,23,24]. Meta-analysis showed that F-FDG PET/CT had a moderate accuracy for the early prediction for pCR (sensitivity 85%, specificity 79%) [25]. DCE-MRI proved to be an effective method for early monitoring the efficacy during NAT (sensitivity 87%, specificity 82%) [26]. However, more authors suggested that MRI studies are heterogeneous with confounders and technical variations in MRI accuracy not adequately studied, so the value of DCE-MRI for response evaluation needs to be further established [26,27].

Similar to our results, tumor size showed no difference before treatment between responders and non-responders (*p*-value > 0.05) [27].

ER and PR negative and HER2 positive status were associated with early NAT re-sponse assessed by US [23]. In our study, ER, PR, HER2 status or ki67% proliferation index were not associated with responders (all *p*-values > 0.05).

Conflicting results have been reported for ADC values, with studies showing a lower ADC mean for responders (cut-off value of 1.17 × 10^−3^ mm^2^/s, *p*-value—0.004), and one study reporting no differences between responders and non-responders [27,28,29].

MR relaxometry proved to be useful in differentiating benign from malignant breast tumors, but its role in assessing NAT response was not investigated. We found that one relaxometry parameter (T2min) was associated with early response (*p*-value = 0.017) and achieved an AUC of 0.715 in predicting NAT response. MR relaxometry reached higher sensitivity (88% vs. 80–87%) and lower specificity (82% vs. 69.2%) compared to previously reported DCE-MR data [26]. Further studies could evaluate the role of MR-DCE together with MR relaxometry in predicting early response to treatment.

Furthermore, N stage together with the T2min relaxometry parameter achieved a higher AUC of 0.826 [95% CI: 0.66–0.90, *p*-value < 0.001], without reaching a difference of statistical significance (*p*-value = 0.34).

The current study has some limitations: (1) small number of patients; (2) other neo-adjuvant therapies such as cyclin-dependent kinase 4 and 6 (CDK4 and CDK6) inhibitors and immunotherapy have little or no published literature on response patterns at breast MRI and were not included in the present study. (3) MR-DCE features (such as internal enhancement type or kinetic curves) were not included in the present study; data are available and will be evaluated on a larger number of patients in a future study.

## 5. Conclusions

ER/PR, HER2 status and the ki67% proliferation index did not differ between early responders and non-responders after NAT. Morphological (size, margins) and functional (ADC values, non-mass component) MR characteristics were not associated with early NAT response. The T2min relaxometry parameter and N stage were associated with early response and can be used to predict NAT response.

## Figures and Tables

**Figure 1 cancers-14-05866-f001:**
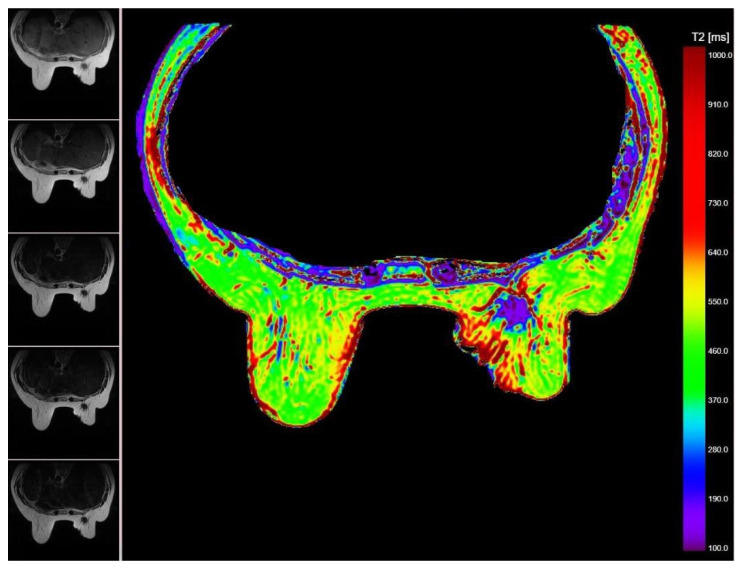
Breast MR relaxometry map (right image) showing a right sided breast carcinoma with different relaxometry values (purple; <190 ms) compared to adjacent breast tissue. The 5 different MR images used to obtain the relaxometry map are displayed on the left side.

**Figure 2 cancers-14-05866-f002:**
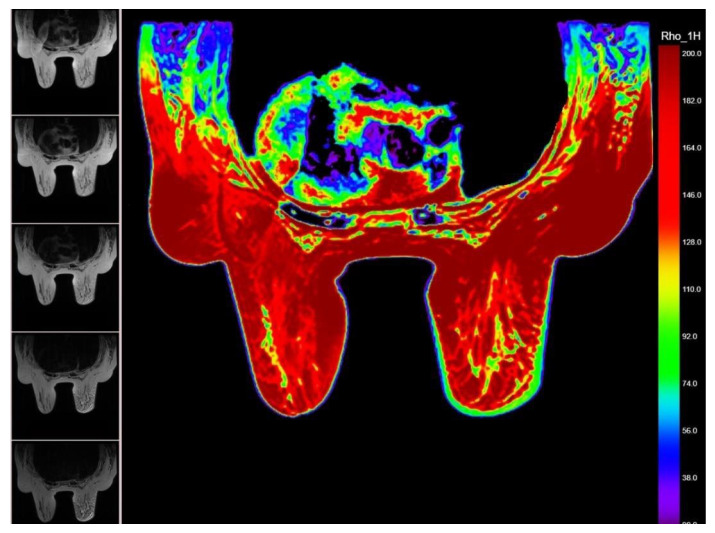
MR proton density map showing carcinomatous mastitis of the right breast with thick-ened skin and malignant interstitial edema (green; between 92–74 ms). The 5 different MR images used to obtain the relaxometry map are displayed on the left side.

**Figure 3 cancers-14-05866-f003:**
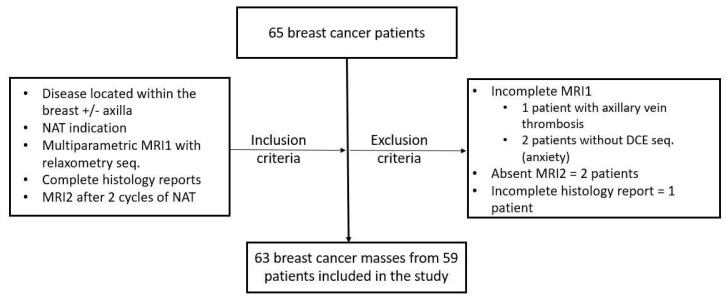
Study population-inclusion and exclusion criteria.

**Figure 4 cancers-14-05866-f004:**
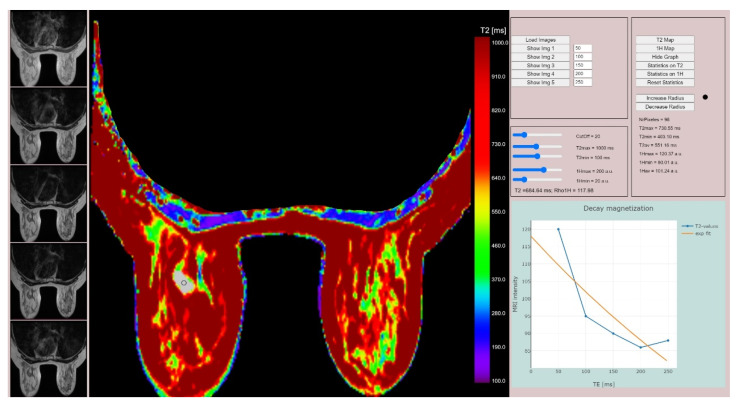
MR relaxometry map–Extracting relaxometry data. The left sided breast tumor is man ually delineated on relaxometry map (grey area), providing the decay magnetization curve (lower right side of the image).

**Figure 5 cancers-14-05866-f005:**
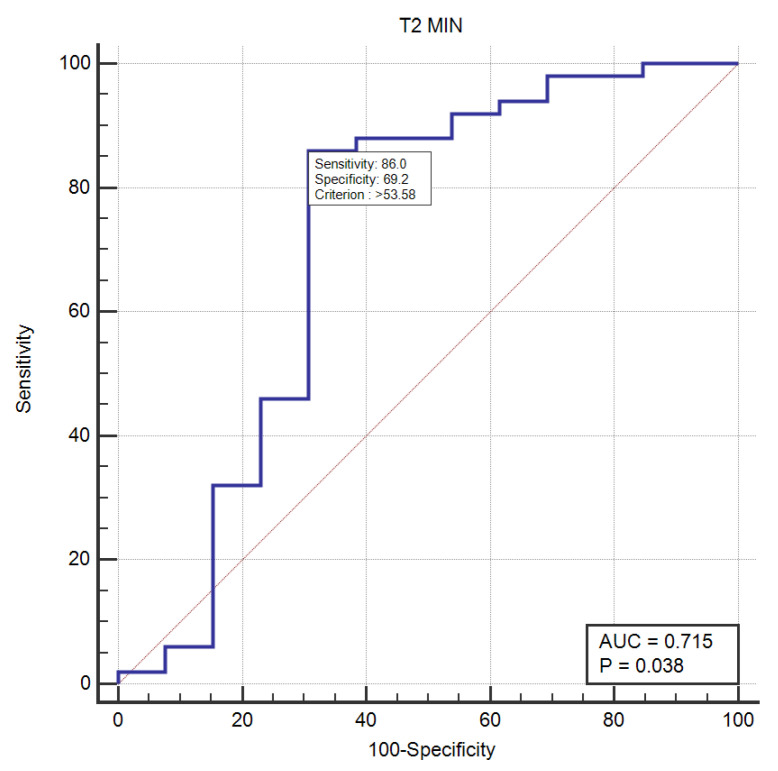
ROC curve for T2min relaxometry parameter in predicting early treatment response to NAT.

**Figure 6 cancers-14-05866-f006:**
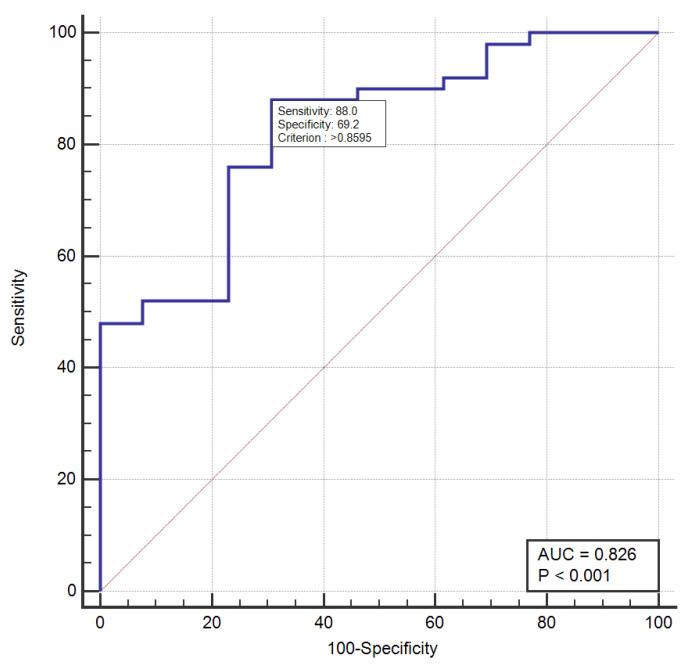
ROC curve for combined model based on T2min relaxometry parameter and N stage, in predicting early treatment response to NAT.

**Figure 7 cancers-14-05866-f007:**
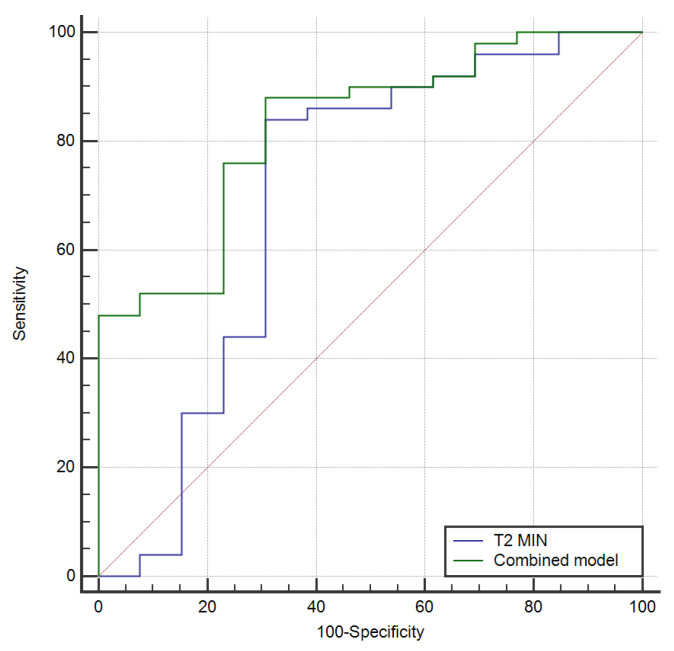
ROC curves comparison between T2min relaxometry parameter and combined model in predicting early treatment response to NAT.

**Table 1 cancers-14-05866-t001:** Clinical, pathological and immunocytochemistry characteristics of responder and non-re-sponder patients.

Variable	Responders	Non-Responders	*p*-Value
**Age** (y), median (range)	65 (44–71)	58 (43–75)	0.237
**Breast cancer type**			
IDC–NST	44	11	0.52
Other *	6	2	
“In-situ” component	13	2	0.34
**Number of tumors**			0.13
Unifocal	32	2	
Multifocal/Multicentric	18	11	
Bilateral	3	1	0.82
**Size mean (mm)**	30.32	27.69	0.56
**Histologic grade**			0.54
Low–G1	11	5	
Intermediate–G2	27	5	
High–G3	10	3	
**Immunohistochemistry**			
ER +	40	11	0.52
ER –	10	2	
PR+	35	9	0.60
PR-	15	4	
HER2+	13	1	0.148
HER2-	37	12	
**Ki-67% status**			
>20%	30	6	0.278
<20%	20	7	
**TNM stage**			
T stage			0.96
1	8	1	
2	31	7	
3	6	1	
4	5	4	
N stage 0			**0.005**
1	25	3	
2	19	3	
3	5	7	
	1	0	
**Total**	50	13	

* Other = invasive lobular carcinoma, mucinous, metaplastic and papillary carcinoma; ER = estro-gen receptor; PR = progesterone receptor; HER2 = Herceptin 2 receptor.

**Table 2 cancers-14-05866-t002:** MR relaxometry features of responders and non-responder patients.

Variable	Responders	Non-Responders	*p*-Value
**Tumor location** * UOQ UIQ LOQ LIQ	28	7	
	12	4	
	10	2	0.69
	NA	NA	
**Mass-Margins**			
Circumscribed	10	2	0.528
Non-Circumscribed	40	11	
**Non-mass**			
Present Absent	13	2	0.641
	37	11	
**ADC mean**	1.02	0.89	0.872
**Relaxometry** Pixel NR T2max	1552.72	1456.77	0.476
	91,790.9	207,447.9	0.316
T2min T2av 1H max 1H min 1H av	975.4	93.66	**0.017**
	452.8	400.2	0.622
	392.4	513.5	0.228
	946.8	3491.1	0.865
	132.9	139.5	0.445

* UOQ = upper-outer quadrant; UIQ = upper-inner quadrant; LOQ = lower-outer quadrant; LIQ = lower-inner quadrant; pixel NR = number of pixels corresponding to the delineated tumor mass; max = maximum value; min = minimum value; av = average values; 1H = hydrogen proton density; NA = no cases.

## Data Availability

Not applicable.

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
