# Peer review of "Predicting the Early Response to Neoadjuvant Therapy with Breast MR Morphological, Functional and Relaxometry Features—A Pilot Study"

_cancers, 2022, doi:10.3390/cancers14235866_

Round 1

Reviewer 1 Report

Dear authors, I would recommend to correlate the findings with particular intrinsic suybtypes - e.g. luminal A,B, TNBC....

Figure 4 needs some corrections - margins are not smooth

Despite my remarks, the study is well written and after some minor revisions the paper could be accepted for publication

Reviewer 2 Report

Dear authors, I had the opportunity to read your interesting manuscript. Maybe it is possible to condense the single references into one larger table? Kind regards 

Reviewer 3 Report

In subgroup analysis the samples size has reduced significantly. Did the authors apply any corrections to the p-value? 

The age of the patients ranged from 41 to 71 which is very large difference. Sub-group analysis based on pre-menopausal and post-menopausal status will add more significance to the data. 

Did the authors consider analyzing triple negative or triple positive breast cancer groups?

Line 140, 142-144 which two groups are you referring here?

Line 138 "there were" should be replaced with there was.

Line 149 "Statistical significance association" replace with statistically significant association

Grammer editing is required.

Round 2

Reviewer 3 Report

The manuscript has been revised according to the comments.